# Various ways of using Empirical Orthogonal Functions for Climate Model evaluation

Rasmus E. Benestad[1], Abdelkader Mezghani[1], Julia Lutz[1], Andreas Dobler[1], Kajsa M. Parding[1], and Oskar A. Landgren[1]

[1]The Norwegian Meteorological Institute, PO Box 43 Blindern, 0313 Oslo, Norway

**Correspondence:** Rasmus Benestad (rasmus.benestad@met.no)

**Abstract.** We present a framework for evaluating multi-model ensembles based on common empirical orthogonal functions ('common EOFs') that emphasise salient features connected to spatio-temporal covariance structures embedded in large climate data volumes. This framework enables the extraction of the most pronounced spatial patterns of coherent variability within the joint data set and provides a set of weights for each model in terms of principal components which refer to exactly the same set of spatial patterns of covariance. In other words, common EOFs provide a means for extracting information from large volumes of data. Moreover, they can provide an objective basis for evaluation that can be used to accentuate ensembles more than traditional methods for evaluation, which tend to focus on individual models. Our demonstration of the capability of common EOFs reveals a statistically significant improvement of the sixth generation of the World Climate Research Programme (WCRP) Climate Model Intercomparison Project (CMIP6) simulations over the previous generation (CMIP5) in terms of their ability to reproduce the mean seasonal cycle in air surface temperature, precipitation, and mean sea-level pressure over the Nordic countries. The leading common EOF principal component for annually/seasonally aggregated temperature, precipitation and pressure statistics suggest that their simulated interannual variability is generally consistent with that seen in the ERA5 reanalysis. We also demonstrate how common EOFs can be used to analyse whether CMIP ensembles reproduce the observed historical trends over the historical period 1959–2021, and the results suggest that the trend statistics provided by both CMIP5 RCP4.5 and CMIP6 SSP245 are consistent with observed trends. An interesting finding is also that the leading common EOF principal component for annually/seasonally aggregated statistics seems to be approximately normally distributed, which is useful information about the multi-model ensemble data.

## 1 Introduction

The question of how to evaluate climate models is often complicated by large volumes of data. In many cases, it is the salient information about meteorological phenomena, conditions, and states that they are designed to reproduce that we want to assess, rather than details in individual grid-boxes that are subject to surface parameterisation and numerical algorithms associated with discrete mathematics, approximations, and statistical fluctuations. The climate models are expected to have an intrinsic minimum skillful scale that arises from discrete mathematics, approximations, and parameterisation (Benestad et al., 2008). Furthermore, they are typically used to study trends and variability but are not expected to be directly synchronised or correlated

in time with the particular timing of 'chaotic' meteorological phenomena playing out in earth's climate (Lorenz, 1963), such as the El Niño Southern Oscillation (Philander, 1989) or volcanic eruptions. Hence, one approach for evaluating them may relate to the spatio-temporal covariance structure embedded in the simulated output. An emphasis on the spatio-temporal covariance can also make use of the redundancy in the data and reduce the degrees of freedom in the data, and hence minimise the required data volume needed for describing the output. Empirical orthogonal functions (henceforth 'EOFs') represent a mathematical technique that identifies the spatio-temporal covariance structure, based on linear algebra and eigenfunctions (Lorenz, 1956; Preisendorfer, 1988; Wilks, 2006; Navarra and Simoncini, 2010; Joliffe, 1986; Hannachi, 2022). Their function may also be considered as a way of reorganising the information embedded within a data object $X$ according to the decomposition

$$X = U\Lambda V^T, \tag{1}$$

where information associated with salient covariance structures is "moved to the top of the list". In this case, we distinguish between the concepts of *data* (mere a large set of numbers) and *information* (what the data represents or its statistical/mathematical properties). The matrix $X$ contains a joint dataset and is $X = [X_1, X_2, \cdots X_n]$ and the principal components in the matrix denoted by symbol $V$ can be expressed as $V = [V_1, V_2, \cdots V_n]$. The first segments $X_1$ and $V_1$ typically hold data from a reanalysis and the others contain data from climate models. Moreover, the components of an EOF analysis have useful mathematical properties, where $U^T U = V^T V = I$ are an identity matrix and $\Lambda$ is a diagonal matrix related to the eigenvalues. The technique may be regarded as a form of machine learning (ML) where EOFs are based on eigenfunctions and eigenvectors for which their mathematical properties simplify the analysis of the data (Wilks, 2006).

Flury (1984); Flury and Gautschi (1986); Sengupta and Boyle (1993, 1998) and Barnett (1999) proposed a variant of EOFs that they described as *common Empirical Orthogonal functions* (henceforth 'common EOFs') for model intercomparison. The common EOFs are mathematically identical to ordinary EOFs, but involve two or more datasets combined on a common grid along the time axis. Hence, one segment of the time axis may represent reanalysis data, whereas another segment may contain climate model data that have been interpolated onto the same grid as the reanalysis. Keeping track of which time segment represents which dataset (in this case a reanalysis or a particular climate model) is essential for common EOFs to make sense. Benestad et al. (2008) described common EOFs and discussed their application in climate research, and common EOFs have been useful as a framework for empirical-statistical downscaling (Benestad et al., 2001), motivated by Barnett (1999). Benestad et al. (2017, 2019a) also provided demonstrations of how common EOFs can be applied to analyse ensembles of decadal forecasts. However, a general literature search with Google Scholar, the assessment reports of the Intergovernmental Panel on Climate Change (IPCC), and the documentation behind the ESMValTool (Eyring et al., 2020; Weigel et al., 2021) suggests that common EOFs are not widely used in the climate research community. The impression of a modest interest in common EOFs was also expressed in Benestad (2021) and is supported by a quote from Hannachi et al. (2022): "To the best of our knowledge only two studies considered common EOFs, which go back more than two decades (Frankignoul et al., 1995; Sengupta and Boyle, 1998), which were based on the original Flury and Gautschi (1986) (FG86)'s algorithm". However, we also know of a few additional cases where common EOFs were employed, e.g. those cited above, that were overlooked by Hannachi et al.

(2022). A Google Scholar search for "common principal component analysis" (1,680 hits, 2022-12-01) nevertheless suggests that common EOFs are discussed in scientific journals belonging to other scientific disciplines than climate science, such as statistics, biometrics, biology geology, and neuro-computing.

To demonstrate the merit of common EOFs, we present some examples of how they can be used to evaluate climate models in the context of large multi-model ensembles of global climate models (GCMs). One of our objectives was to evaluate GCM data that are used as predictors for empirical-statistical downscaling (ESD) over Northern Europe, and we picked this as an example to demonstrate their utility and merit. While Hannachi et al. (2022) used common EOFs to compare individual models, we present an approach where they were used to compare different ensembles, such as CMIP5 RCP4.5 and CMIP6 SSP245, and to assess if they provide a statistical representation that has a similar *statistical population* (Wilks, 2006, e.g. p.72) as the reanalysis. In this case, they provided a framework in which we applied standard hypothesis testing and statistical tests.

## 2 Data & Method

Hannachi et al. (2022) provide a description of the mathematics behind common EOFs which also is relevant for our analysis, but here we present a slightly different approach for applying them for the purpose of climate model evaluation and for assessing different model ensembles. Our method bears both similarities and differences to previous applications of common EOFs. In our case, we applied EOFs to one variable from different sources stacked in space-direction as in Barnett (1999), which is similar to a tensor decomposition where the space-time matrices of the individual projections are stacked along a third (model) direction (Cichocki et al., 2015). Our approach with data stacked from different models along the time direction had a similar function as a 'third model dimension' because spatio-temporal covariance matrices only involve time and space. It is also possible to stack data along the space axis, as multi-variate EOFs described in (Sanderson et al., 2015) where all monthly values of any one projection were stacked in space-direction instead of time-direction, however, this is only meaningful if the data are expected to contain synchronous temporal variations. Another example of using such 'mixed field' EOFs is found in Benestad et al. (2002) who combined standardised surface air temperature and mean sea-level pressure and used EOFs of these combined fields as predictors in empirical-statistical downscaling. The different CMIP runs are not expected to be synchronised as the regional variations are both pronounced and of chaotic-stochastic nature.

We used singular value decomposition (SVD) (Becker et al., 1988) on a joint data matrix (multiple data sets stacked along the temporal axis) rather than the step-wise algorithm for a set of covariance matrices described by Hannachi et al. (2022). Hence, we obtained identical spatial maps and eigenvalues for all models in the joint data matrix, but different statistical properties (e.g. amplitude and mean) for the different segments of the principal components that represented different models. Both these variants have been referred to as "common EOFs", and for the lack of a better term, we will use the term "common EOFs" for the analytical framework presented herein. In our case, we used the approach described in Benestad et al. (2019a) where common EOFs were used to represent an ensemble of decadal forecasts based on a single GCM. More specifically, we used common EOFs to illustrate how well GCMs reproduce the mean annual cycle in terms of the spatio-temporal covariance structure, compared with the ERA5 reanalysis (Hersbach et al., 2020). We also present another example where we used common

EOFs to assess how well the GCMs simulate the interannual variability in terms of the annual mean surface air temperature, precipitation, and mean sea-level pressure. A third way of applying EOFs in model evaluation is as a framework for comparing trends simulated by different GCMs, where they highlight salient features in the trend structure. In all these cases, we used common EOFs to evaluate both CMIP5 (Meehl et al., 2005; Taylor et al., 2012) and CMIP6 (Eyring et al., 2016) ensembles in a joint analysis. One complication was the varying number of simulations carried out with one model set-up, as some GCMs had produced numerous simulations in the CMIP ensembles whereas others had only produced a few. To make the evaluation as objective as possible, we only selected one simulation from each GCM, filtering the data based on the ensemble member label ($r1i1p1$ for CMIP5, $r1i1p1f1$ for CMIP6), using runs that spanned the period 1850–2100, and only the emission scenarios RCP4.5 and SSP245 (in this case, we only used data for the common period with the ERA5 reanalysis: 1959–2021). We also repeated the analysis on slightly different spatial domains to assess the robustness of our results. In this evaluation, we computed common EOFs for a joint dataset of 35 CMIP5 RCP4.5 runs and 40 CMIP6 SSP245 runs (75 runs in total for TAS, but not all of these were available for PR and PSL). To cope with the vast amount of data, each model run was represented in terms of monthly mean seasonal cycle (12 calendar months each) as well as annually/seasonally aggregated statistics (63 spatial maps for each run, one for each year in the period 1959–2021). In this case, the term *aggregated statistics* refers to the mean estimate for TAS and PSL and the sum for PR (total precipitation over the year or season).

Common EOFs can be used to evaluate individual GCMs against the ERA5 reanalysis through the estimation of difference in the mean $\overline{x}$, standard deviation $\sigma$, and lag-1 autocorrelation $r_1$ estimated for the different segments of the principal components (PCs) representing different datasets (Benestad et al., 2016, 5.1 in the Supplementary data). Here, we evaluated how well the models are able to reproduce the mean seasonal cycle in the surface air temperature (TAS), monthly precipitation totals (PR), and the mean sea-level pressure (PSL) over a region spanning the Nordic countries: 5°W–45°E and 55–72°N. We repeated the same analysis in four other domains to assess the robustness of our results and those findings can be found in the supporting material available from FigShare (Benestad, 2022). The model performance was gauged by taking the root-mean-square error (RMSE) of the leading principal component that accounts for most of the variance, using ERA5 as a reference. We also applied common EOFs to the annual/seasonal mean TAS, annual/seasonal total PR, and annual/seasonal mean PSL to diagnose their interannual variability and how well it was reproduced in the CMIP ensembles. Moreover, we applied the analysis separately to the full year (Jan-Dec) and each of the four seasons: winter (DJF), spring (MAM), summer (JJA), and autumn (SON). The skill metric of the models' reproduction of the interannual variation in the said annually/seasonally aggregated statistics involved rank-statistics and the assumption that any rank is equally probable if the weights of the PC representing the ERA5 reanalysis belongs to the same statistical population as the ensemble of GCMs. We used a two-sided Kolmogorov-Smirnov test (Wilks, 2006) to compare the empirical distribution of the rank-statistics against a uniform distribution representing the case for which all ranks have the same probability. We also used Monte-Carlo simulations to represent a 'perfect case' as a reference for the rank analysis of the annual/seasonal means. In these simulations, we used the same number of years and a statistical sample with the same size as the ensemble in question and picked a fixed realisation as a 'surrogate' for the 'reanalysis' and the rest to represent the 'ensemble'. In these Monte-Carlo simulations, the 'reanalysis' and 'ensemble' belonged to the same statistical population by design.

Finally, we made a data matrix with columns consisting of spatial maps (the 2D matrix orientation of the data was reordered into a 1D vector) with linear trend estimates over 1959–2021 with one column for each GCM in the respective CMIP multi-model ensemble, in addition to a corresponding map with trend estimates derived from the ERA5 reanalysis. The EOFs of this joint data matrix were used to assess the differences in reproducing the main aspects of the historical trends among the GCMs and reanalysis.

The analysis presented here was carried out using the R-package 'esd' version 1.10.15 (Benestad et al., 2015) within the R-environment version 4.2.2 (R Core Team, 2014). Essential data and R-code (an R-markdown script and its output in PDF format) used for these computations are available as supporting material and as free open-source material from FigShare in order to enhance the transparency and reproducibility of these results: https://figshare.com/articles/dataset/Common_EOFs_ for_model_evaluation/21641756. The FigShare repository can be cited as Benestad (2022) and is archived as a combination of the R-markdown script, the PDF file (supporting material for this paper), and a set of R-binary data files stored as separate files for the respective RCP45 and SSP245 scenarios and for the three different parameters TAS, PR, and PSL. The data files contain 72-75 different GCM runs in addition to ERA5, and the total data volume of all these files is 1.9 GB. While the processing of the data stored in this repository was carried out on powerful Linux servers and the job for all combinations of seasons and regions took roughly 22 hrs to complete, the R-code provided was run on a 64-bit HP Elitebook 850 G8 laptop with Ubuntu 18.04.6 LTS with 32 Gb memory.

It is possible to test the common EOF framework for cases with 'bad' data to see how the results turn out for when the ensemble does not reproduce the properties of the reference. In this case, we simulated such a case by replacing the ERA5 TAS with ERA5 PR keeping the ensemble as it were (TAS) so that the reference and ensemble consisted of different types of variables (supporting material). The mismatch could be seen in the amplitude of the PCs representing the reference and ensemble as well as in the leading EOFs representing a lower fraction of the variance. The EOFs were dominated by the ensemble and in general it may be necessary to include several PCs in the evaluation when the leading EOFs do not account for most of the variance. It's also important to keep in mind that the PCs' variance fractions may depend on the spatial domain covered by the data grid.

## 2.1 Results

### 2.1.1 Evaluation of the simulated mean seasonal cycle

Figure 1 presents the leading common EOF for the mean seasonal cycle in the surface air temperature (TAS) over the Nordic countries. The spatial map (upper left panel) shows the structure of the most dominant covariance pattern of the seasonal cycle, and the eigenvalues (upper right panel) suggest that this mode dominates the seasonal behaviour completely. Both pattern and eigenvalues were estimated from the joint dataset that involved ERA5, the CMIP5 RCP4.5 ensemble, and the CMIP6 SSP245 ensemble. The spatial patterns ($U$ in equation 1 shown in the upper left panel) and the eigenvalues ($\Lambda$ in equation 1 presented in the upper right panel) are common for all models, and only the corresponding principal components (PCs, represented by the matrix $V$ in equation 1) in the lower panel show differences between the reanalysis and the GCMs from the CMIP5 and

CMIP6 ensembles. These differences are visible as scattered brown and green curves. It is important to keep in mind that individual EOFs may not necessarily be associated with a clear physical meaning, especially the higher order ones, as the different modes are designed to be orthogonal to each other (Ambaum et al., 2001; Huth and Beranová, 2021). However, they are useful mathematical concepts that enable more efficient work with large data volumes and make it easier to extract salient information from it, but sometimes they nevertheless may provide insights on physical phenomena within the analysed domain. In our analysis, they ensured a set of indices for all GCMs which were related to a common covariance structure within the joint dataset, and we used them to evaluate the mean seasonal cycle estimated over the period 1959–2021. Our evaluation was based on the root-mean-square error (RMSE) between the leading PC representing the corresponding mean seasonal cycle in TAS from ERA5 and the joint set of 75 GCMs from both CMIP5 RCP4.5 (35 members) and CMIP6 SSP245 GCMs (40 members). The results of this evaluation are presented in Table 1 and a Wilcoxon Rank Sum (also known as Mann-Whitney) test (Wilks, 2006) was applied to the two sets of RMSE scores representing CMIP5 and CMIP6 respectively. Our results indicated that the CMIP6 simulations had a better score, and the difference with CMIP5 was statistically significant at the 5% confidence level. Hence, the CMIP6 models were more skilful at reproducing the mean seasonal cycle in TAS in the Nordic region. The difference in skill is also visible in the lower panel, which shows that the curves for CMIP5 (brown) were less tightly clustered around ERA5 (black) than those for CMIP6 (green). The leading mode accounted for 96% of the variance, which suggests that all GCMs produced a seasonal cycle with a similar spatial covariance structure (upper left).

We repeated the evaluation of the climate models' ability to reproduce the mean seasonal cycle in PR (Figure 2 and Table 2) and PSL (Figure 3 and Table 3). The number of available CMIP results for PR was slightly different to that of TAS at the time of the analysis, and our ensembles consisted of 33 members from CMIP5 RCP4.5 and 37 from CMIP6 SSP245. The exact ensemble size wasn't critical for our demonstration, as our objective was to demonstrate the utility and merit of common EOFs for model evaluation. The eigenvalues for PR indicated that the leading mode accounted for a lower portion of the variance (71%) than TAS, which may be due to variations in their ability to capture the typical spatial patterns in PR associated with different seasons. The greatest seasonal variations in PR can be seen near the west coast of Norway (upper left panel of Figure 2). The leading mode for PSL, on the other hand, accounted for 86% of the variance, and most GCMs reproduced a mean seasonal cycle that involved a northwest-southeast PSL gradient. The common EOFs for PSL were applied to 35 members from CMIP5 RCP4.5 and 37 from CMIP6 SSP245. The RMSE scores for PR and PSL are reported in Tables 2–3 and a Wilcoxon rank sum test indicated that the CMIP6 simulations constituted an improvement over those from the CMIP5 in terms of reproducing the mean seasonal cycle, using the ERA5 reanalysis as a reference (statistically significant at the 5%-level).

### 2.1.2 Evaluation of the simulated interannual variability

The results of the evaluation of the interannual variability in the annual mean TAS are shown in Figure 4 in terms of the leading common EOF with a map of the covariance connected to its interannual variability (upper left), eigenvalues (upper right), and time evolution (lower). One striking observation is that the leading mode accounted for 65% of the variance with the five leading modes accounting for approximately 90%, suggesting that most GCMs reproduced a similar covariance structure. We

used a rank metric $\mathcal{R}$ where the PC weights for ERA5 were compared with the spread of the CMIP5 and CMIP6 ensembles in terms of their rank within each year and each ensemble. For the leading mode of the annual mean temperature shown in Figure 4, both the CMIP5 and the CMIP6 produced ensemble results with a statistical population that was likely consistent with ERA5 data. In both cases, the two-sided Kolmogorov-Smirnov test indicated a high probability ('p-value') for $\mathcal{R}$ belonging to a uniform distribution. A p-value close to zero means that the data connected to the part of the leading PC representing ERA5 most likely belonged to a different statistical population than the respective CMIP ensemble (data from different segments of the same leading PC), whereas a p-value near unity implies that ERA5 and the CMIP ensemble more likely belonged to the same statistical population. In our analysis, the Kolmogorov–Smirnov test for CMIP5 returned $D = 0.099206$ with a p-value of 0.5647 and the CMIP6 $\mathcal{R}$ obtained $D = 0.11362$ with p-value $= 0.3902$. Figure 5 provides a visualisation of the rank metric $\mathcal{R}$ on a year-by-year basis (upper panel) as well as a histogram of the ranks for TAS results shown in Figure 4. It is evident from these plots that $\mathcal{R}$ varies over the whole interval $[0, 1]$ and follows a distribution that is more or less uniform ('flat' structure) which we expect for $\mathcal{R}$ if each rank is equally probable. Hence, for the annual mean TAS over the Nordic regions, both CMIP ensembles provided an approximate representation of the interannual variability seen in ERA5 and connected to the leading mode. A set of Monte-Carlo simulations indicated that the ranking scores would fluctuate even with ensembles that mimicked perfectly the statistical properties of the observations, due to the limited sample size.

A corresponding assessment of the leading common EOF for PR (Figure 6) indicated similar differences in statistical terms for both CMIP5 (D = 0.11858, p-value = 0.3384) and CMIP6 (D = 0.13085, p-value = 0.2309). The leading common EOF for annual PR representing variations along the west coast of Norway only accounted for 27% of the variance, but the five leading modes accounted for approximately 60%, suggesting that interannual variability in precipitation involves more complicated anomalies and perhaps greater model differences. The low variance associated with the leading modes may suggest that the models produce different spatio-temporal covariance structures, i.e. that they produce different typical patterns of rainfall. It is also possible that a lower fraction of variance represented by the leading mode is due to smaller-scale spatial structures relative to the domain size when it comes to precipitation patterns. Hence, the common EOFs applied to annual PR revealed a more complicated situation where more than one mode dominates. In this case, the first five PCs represented the most salient properties in terms of PR spatio-temporal covariance, representing more then 60% of variance, and the high-order PCs were associated with negligible variance that typically represent "numeric noise".

Our assessment of how well the GCMs reproduced interannual variations in the annual mean PSL gave similar results as for TAS and PR. The leading mode was characterised by a centre of action over northern Scandinavia and accounted for 62% of the variance (Figure 7). The second and third modes were less important, but the fact that they had similar eigenvalues (16%) suggests that they were "degenerate" which refers to the two patterns being two aspects of the same mode (Wilks, 2006, p.488). For PSL, $\mathcal{R}$ was close to having a uniform distribution and the test did not indicate a difference that was statistically significant at the 5%-level for either CMIP5 (D = 0.13492, p-value = 0.2016) nor CMIP6 (D = 0.14329, p-value = 0.1504). In other words, both CMIP5 and CMIP6 seemed to roughly reproduce the annual mean circulation patterns over the Nordic region seen in the ERA5 data represented by the leading mode. The three leading modes accounted for approximately 94% of the variance, suggesting that the GCMs reproduced a similar covariance structure albeit with slight variations.

We examined the nature of the multi-model ensemble distribution of the data of the leading PC for the annual aggregated statistics for the year 2022 and found them to be approximately normally distributed (Figure 8) for most cases, both when it came to annual and seasonal time scales and for both CMIP5 RCP4.5 and CMIP6 SSP245. Only in a few cases did the data deviate substantially from the diagonal in the Q-Q plot, such as for the annual mean TAS (Figure 8(b)), but this was not the typical outcome (supporting material, Benestad (2022)).

### 2.1.3 Evaluation of the simulated historic trends

Figure 9 shows common EOFs that have been used to compare 1959–2021 trend maps from CMIP5 RCP4.5 (red curve in the lower panel) and CMIP6 SSP245 (blue curve in the lower panel) with ERA5 (black symbol), in this case over the Barents Sea region. Each ensemble member was represented by only one weight in the leading PC. The leading mode dominated by accounting for 94% of the variance, suggesting that all models reproduced patterns with the strongest response in the northeast and weakest in the southwest, albeit with different amplitudes. The CMIP6 SSP245 (blue curve) indicated stronger variability between models than the CMIP5 RCP4.5 (red curve), suggesting a wider range of outcomes for the former and that the CMIP6 ensemble contained some more 'extreme' models. It is nevertheless evident that the spread in both CMIP5 and CMIP6 embraced the results obtained with ERA5.

### 2.1.4 Assessment of robustness

The analyses of the mean seasonal cycle, the interannual variability, and historic trends were repeated for the said aggregated statistics for each of winter, spring, summer and autumn seasons (supporting material) as one motivation behind this evaluation was to assess typical predictors used in empirical-statistical downscaling which mainly involve seasonally aggregated statistics. We obtained similar results for the four different seasons (winter, spring, summer, and autumn). Moreover, the spatial domain (region) was chosen for the benefit of assessing the models before using them as input in downscaling exercises. The use of common EOFs as a framework for downscaling also provides a quality assessment (Benestad, 2001; Benestad et al., 2016), but extending them to larger multi-model ensembles provides a more comprehensive assessment of the entire ensemble. The repeated analysis for different spatial domains gave similar conclusions as those presented here for the 5°W–45°E and 55–72°N domain and the Barents Sea region.

## 2.2 Discussion

Linear algebra, eigenfunctions, and EOFs are well-established and versatile mathematical concepts, but we argue that there still are innovative ways of applying them in data analysis. In these demonstrations, they provided the basis for a framework, referred to as 'common EOFs' that enabled simple data comparisons with an emphasis on the most salient features in the data. It is in general possible that the reference does not fall inside the ensemble spread for individual PCs, for which common EOFs would give low fractional variances for the leading modes and different amplitudes in the PCs. In a way, one could refer to the

application of common EOFs as a kind of machine learning (ML) approach to "Big data", characterised by large data volumes, diverse sources, and speedy analysis.

Our demonstrations revealed a spread in the CMIP GCM ensembles that appeared to be consistent with the ERA5 interannual variability and a spread that often was close to normally distributed. The different ensemble members were independent of each other and could be considered as 'random' in terms of their phase and timing, making the ensemble suitable for representing the non-deterministic natural variability. From a physical point of view, we know that these models reproduce chaotic and stochastic variability on decadal scales and this is especially apparent if the ensemble is made up of simulations with one
common model (Deser et al., 2012). For multi-model ensembles, there is also the additional component in terms of model differences. In one respect, we should indeed expect a strong interdependence between climate models since they are built to represent the same physical system, but what we really desire is that the aspects that are not well-established and uncertain should involve different choices/methods so that they also provide a decent sample of the parameter space of unknowns. But in practice, different groups often copy others' attempts so that model uncertainties are not so well sampled (Boé, 2018).
Nevertheless, the simulated stochastic/chaotic regional internal variability appears to be more pronounced, as both the time series in Figures 4–7 seem to indicate, so these concerns are secondary in this case.

    The independence between each model's representation of random variability together with the ensemble spread being approximately normal suggests that the salient information about future projections can be summarised by two parameters: the ensemble mean $\mu_e$ and the ensemble standard deviation $\sigma_e$. They can provide an estimate of a confidence interval $\mu_e \pm$
$2\sigma_e$, and if the ensemble spread is approximately normally distributed we can use them to project a pdf $\sim \mathcal{N}(\mu_e, \sigma_e^2)$ for future aggregated TAS, PR or PSL statistics on an annual or seasonal basis. This illustrates the difference between data and information, where the collection of time series for all ensemble members constitutes *data* of the ensemble, whereas $\mu_e \pm 2\sigma_e$ provides *information* about the ensemble. Hence, users of regional climate projections may not necessarily need to adapt their analysis to many individual simulations if they can get away with information about potential future outlooks in terms
of a robust confidence interval. A pdf representing the ensemble distribution may also be used as a component of Bayesian inferences to estimate probabilities for e.g. heatwaves or heavy 24-hr precipitation (Benestad et al., 2018, 2019b).

    The common EOF framework also suggested that CMIP6 models were better than CMIP5 models at reproducing the mean seasonal cycle in TAS, PR, and PSL over the Nordic region. Additionally, our analysis proposed that both CMIP5 RCP4.5 and CMIP6 SSP245 multi-model ensembles provide an approximate description of typical predictors on spatial domains relevant
for empirical-statistical downscaling over the Nordic countries. The information about improved simulations in CMIP6 is in line with Lauer et al. (2022) who found that the total cloud cover, cloud water path, and cloud radiative effect, were slightly better in the CMIP6 multi-model mean than the CMIP5 ensemble mean, in terms of mean bias, pattern correlation, and relative root-mean-square deviation. They also noted that an underestimation of cloud cover in stratocumulus regions is still a problem in CMIP6. The clouds simulated by the CMIP5 models were reported to be too few and too reflective over the Southern Ocean,
but were significantly improved in CMIP6.

    The common EOF approach and the esd-tool (Benestad et al., 2015) represent a complement to already existing analysis tools such as the GCMeval tool (Parding et al., 2020) or the Earth System Model Evaluation Tool (ESMValTool) (Eyring et al.,

2020; Weigel et al., 2021). The latter performs common preprocessing operations and diagnostics that include tailored diagnostics and performance metrics for specific scientific applications. It furthermore provides diagnostics for the mean annual cycle, pattern correlation, clustering, and EOFs, with RMSE estimates on a grid-by-grid basis (or spatial means of grid-box estimates) rather than in terms of covariance structure such as in Figure 1 and Table 1. ESMValTool also offers regression of monthly mean geopotential heights onto the leading principal component monthly averaged to represent the Northern Annular Mode (NAM), rather than a common EOF approach similar to that presented in Figure 4. It makes use of the Climate Variability Diagnostics Package (CVDP) that computes key metrics of internal climate variability in a set of user-specific model simulations and observational data sets, providing spatial patterns and time series (Phillips et al., 2014). Although it offers a large collection of diagnostics and performance metrics for atmospheric, oceanic, and terrestrial variables for the mean state, trends, and variability, it does not include common EOFs. While the ESMValTool is designed for the evaluation of climate model performance on a more individual basis rather than how well multi-model ensembles represent the world, the common EOF framework proposed here can be used to assess whether the multi-model ensemble is fit for representing climate change and non-deterministic climate variability. Hence, the common EOF framework can be designed to assess model results with a focus on their application for climate change adaptation. The ESMValTool has been developed as a community effort currently involving more than 40 institutes with a rapidly growing developer and user community. It offers more *predefined* functionalities than the esd-package, but the esd-package is more generic, flexible, and also more geared towards empirical-statistical downscaling (ESD). ESD can also provide diagnostics about GCMs (Benestad, 2021), and the esd-package is designed to deal with a more varied set of data types than just GCM output, as it has evolved from a previous open-source R-library `clim.pact` (Benestad, 2003). Both these tools can likely benefit from closer collaboration than in the past, as they seem to complement each other. Moreover, common EOFs make it easy to avoid matrices of many small maps ("stamp collections") that are difficult to digest, since comparisons can be limited to time series and their differences in terms of statistics. Finally, common EOFs also give a visual impression of simulated quality, as well as a framework for more objective tests when applied to their principal components.

The results presented here for TAS represent a typical "easy" case where the variance is represented overwhelmingly by the first EOF and ERA5 lies well in the middle of the ensemble distribution. The results for PR exhibit a more complicated situation in the interannual variation where higher-order PCs represent a greater fraction of the covariance structure, and in our case, the 20 leading modes merely accounted for about 80% of the variance. For a more complete evaluation, the RMSE score metric $e_m$ needs to include higher-order PCs, and according to Equation 1, we get

$$e_m = \frac{\sqrt{\sum_{i=1}^{N} \Lambda_i^2 \sum_t (V_{m,i,t} - V_{1,i,t})^2}}{N}, \qquad (2)$$

where $N$ is the number of modes, $\Lambda$ is the eigenvector, and $V_{m,i,t}$ is the $i$th PC for model $m$ in the ensemble references that consists of a time series over time period $t$. The leading EOFs are associated with higher fractions of variance and higher eigenvalues $\Lambda_i$, whereas the RMSE is less sensitive to higher-order ones with low $\Lambda_i$.

One choice regarding the use of this approach for evaluation is to include only the ensemble of projections in the EOF analysis, and then to project one or several reference reanalyses onto these patterns. This variation of our approach would represent "cleaner" approach not to meddle projections and references, especially if the projections involve other time periods and future outlooks. However, the exclusion of the reanalysis from estimating the EOFs would hardly make any appreciable difference with such vast ensembles as used here and with the same time coverage. In our case, the hypothesis was that the

selected reanalysis and model data represent the same statistics, variable, region and time period, and hence should have similar properties. It is of course possible that the GCMs and reference differ so much so that the reference is outside the ensemble spread, which would indicate that they belong to different statistical populations. This matters for the first PCs representing a large fraction of variance, but can be ignored for high-order PCs associated with negligible variance that represent "numeric noise".

## 3   Conclusions

We present some demonstrations of how common EOFs can be applied in global climate model evaluation and use them to show that the CMIP6 SSP245 multi-model ensemble represents an improvement over CMIP5 RCP4.5 when it comes to reproducing the mean seasonal cycle in the near-surface temperature, precipitation, and mean sea-level pressure over the Nordic countries. The analysis based on common EOFs also suggests that both CMIP ensembles are able to reproduce interannual variability of these variables over the Nordic region and that they seem to embrace the observed historical trend seen in the ERA5

reanalysis. Common EOFs are not widely used within the climate research community and we propose that they may benefit further research through innovative applications. A motivation for using common EOFs was to assess the value of multi-model ensembles of climate models for the application in climate services, rather than focusing on single models. Hence, they were used to answer the question of whether the said CMIP multi-model ensembles are able to reproduce the observed statistics of

the regional climate that is necessary for supporting climate change adaptation.

*Code and data availability.*   Both R-markdown scripts with embedded R-code, output in the PDF-format and data in R-binary are available from FigShare (Benestad, 2022).

*Video supplement.*   A couple of YouTube demonstrations on common EOFs are available from https://youtu.be/32mtHHAoq6k and https://youtu.be/E01hthVL9pY.

*Author contributions.*   REB conceptualised the work, carried out the analysis, and participated in the writing; KMP and AM have contributed to the write-up and the development of the esd-package used to compute common EOFs and carry out the analysis; JL, AD and OAL contributed to the writing process.

*Competing interests.* None.

*Acknowledgements.* Several datasets (CMIP5 and CMIP6) used in this work were obtained from the CMIP6 project hosted on the Earth
System Grid Federation https://esgf-data.dkrz.de/esg-search/search/ and CMIP5 through the KNMI ClimateExplorer https://climexp.knmi.
nl/start.cgi. The ERA5 reanalysis was obtained through the Copernicus Climate Change Services (C3S) Climate Data Store (CDS): https:
//cds.climate.copernicus.eu/cdsapp#!/dataset/reanalysis-era5-single-levels-monthly-means?tab=form The analysis was implemented in the
R-environment (R Core Team, 2014) and R-studio https://posit.co/downloads/.

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

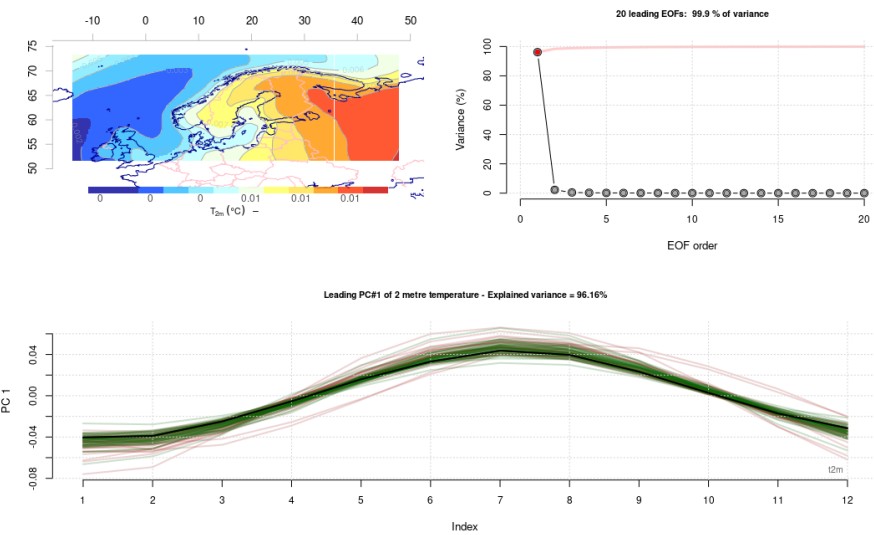

**Figure 1.** Common EOFs which present the covariance structure for model simulations of the annual mean cycle in TAS. The upper left panel presents the spatial covariance structure of the leading mode, the upper right indicates the variance associated with 20 leading modes, and the lower panel shows the leading PC for the multi-model ensemble. The black curve represents the ERA5 reanalysis, whereas the red curves represent CMIP5 and the blue curves CMIP6.

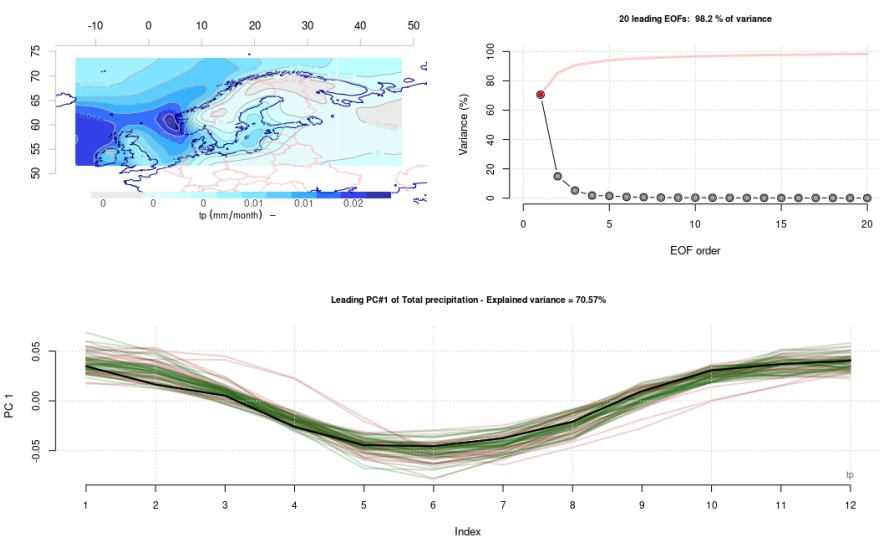

**Figure 2.** Same as 1 but for the mean annual cycle in precipitation.

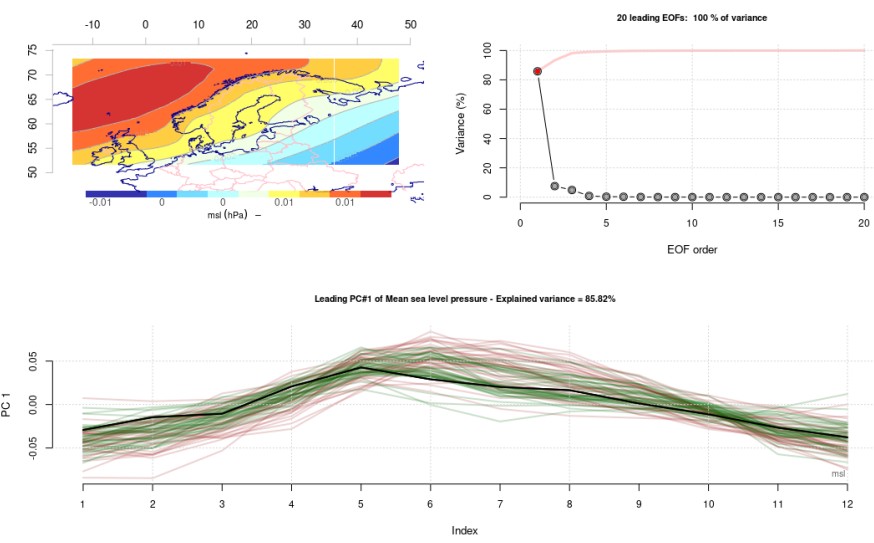

**Figure 3.** Same as 1 but for the mean annual cycle in sea level pressure.

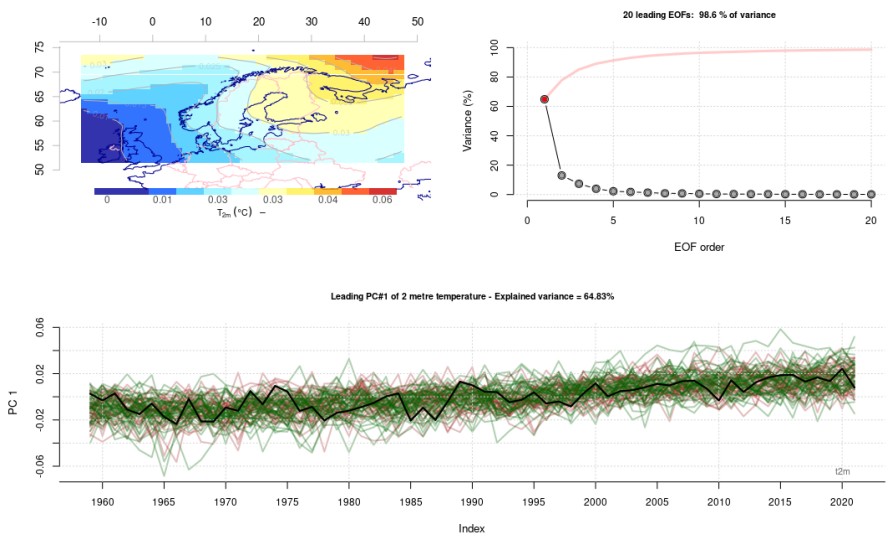

**Figure 4.** Common EOFs which present the covariance structure for model simulations of the interannual variability in the annual mean TAS. The upper left panel presents the spatial covariance structure of the leading mode, the upper left indicates the variance associated with 20 leading modes, and the lower panel shows the leading PC for the multi-model ensemble. The black curve represents the ERA5 reanalysis, whereas the red curves represent CMIP5 and the blue curves CMIP6.

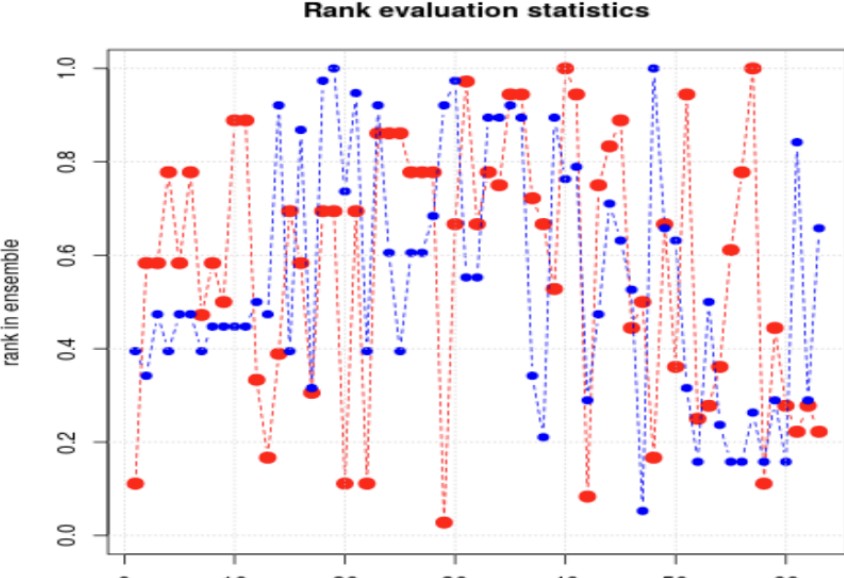

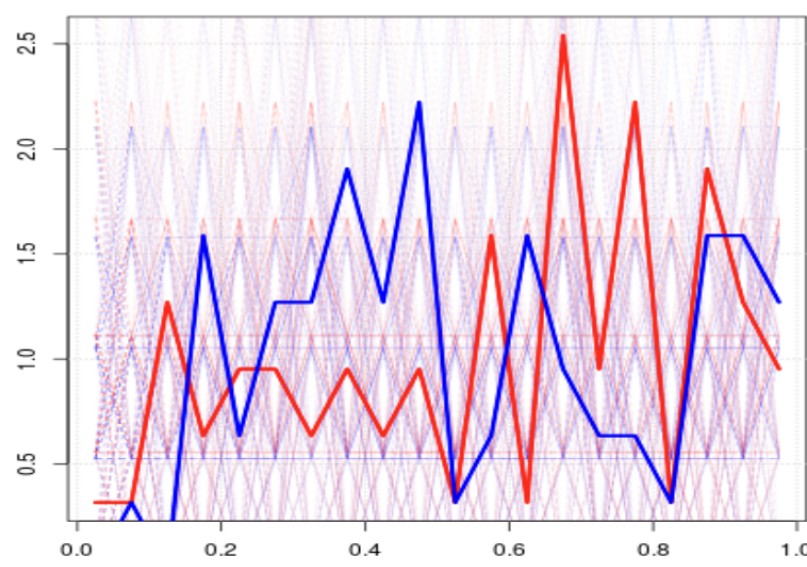

**Figure 5.** Rank statistics $\mathcal{R}$ for the case presented in Figure 4 where the upper panel shows the rank of ERA5 results within the multi-model ensemble spread on a year-to-year basis, whereas the lower panel shows histograms of the rank statistics together with results from Monte-Carlo simulations of perfect cases (y-axis shows frequency and x-axis the range of rank categories). Red marks CMIP5, whereas blue marks CMIP6. Kolmogorov–Smirnov statistic $D$ for CMIP5 was D = 0.099206 with a p-value = 0.5647 and the CMIP6 $\mathcal{R}$ obtained D = 0.11362 with a p-value = 0.3902.

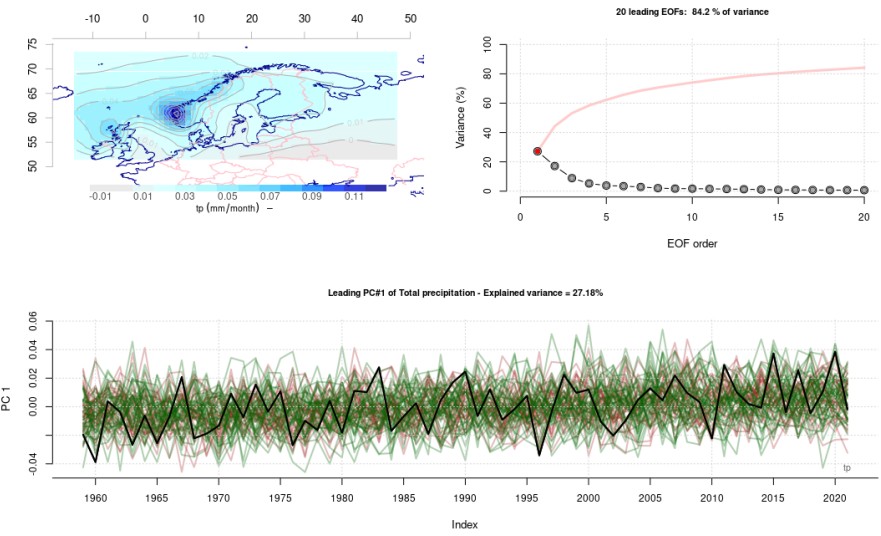

**Figure 6.** Same as 4 but for the annual mean precipitation PR.

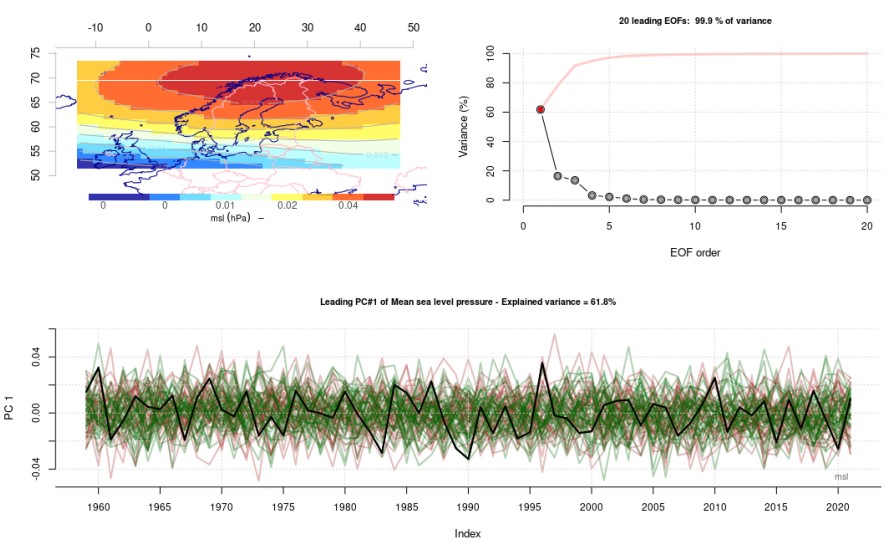

**Figure 7.** Same as 4 but for the annual mean PSL.

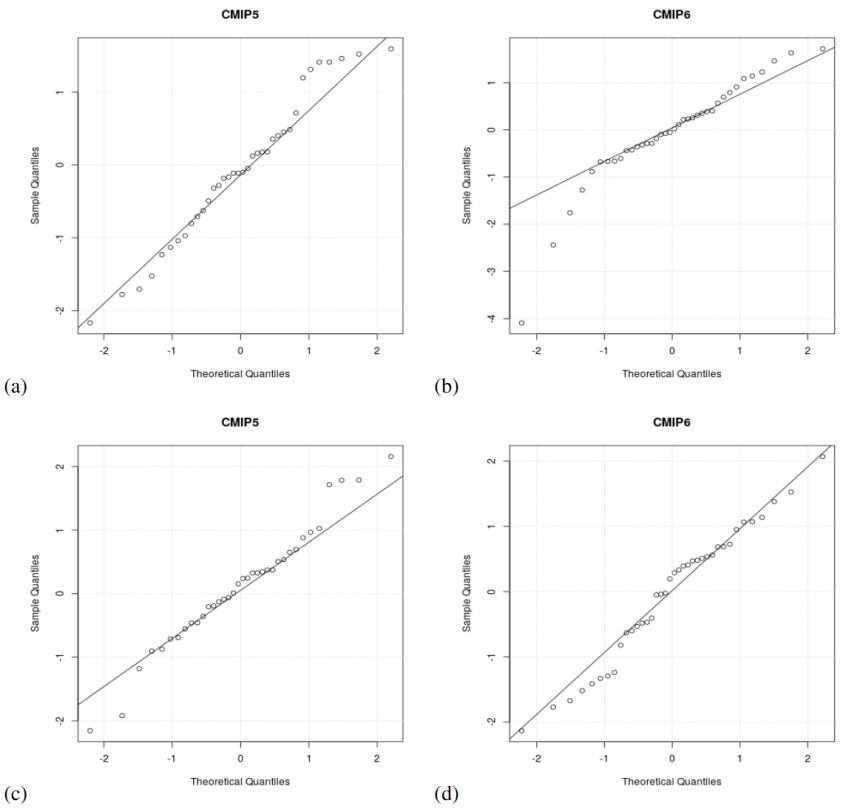

**Figure 8.** Normal Q-Q Plots for CMIP5 RCP4.5 (left panels) and CMIP6 SSP245 (right panels) and for annual mean TAS (upper panels) and winter mean TAS (lower panels) showing variations in the nature of the ensemble distribution. The more pronounced deviation from a normal distribution for annual mean CMIP6 TAS in panel (b) was untypical for these results. Each data point represents the leading PC weight for 2021 for one GCM, i.e. the end points of the time series in the lower panels of Figures 4, 6 and 7.

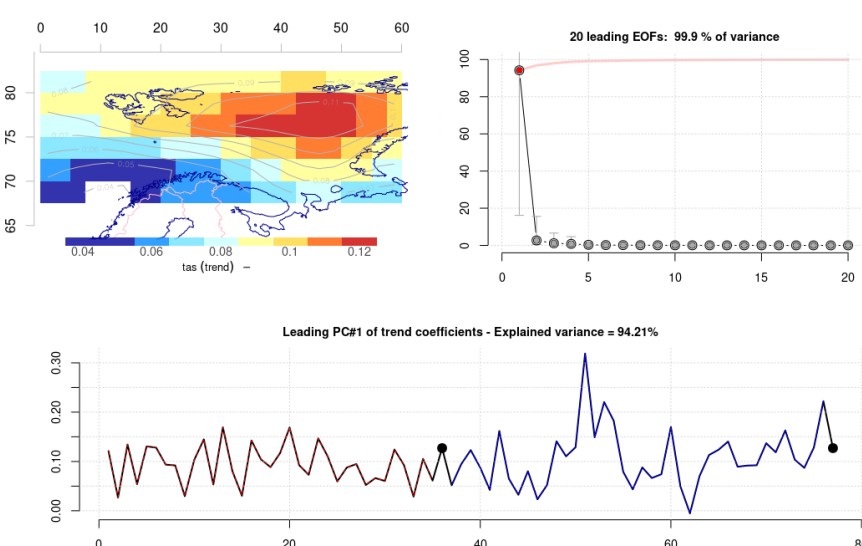

**Figure 9.** Common EOFs which present the covariance structure for model simulations of trend maps in the annual mean TAS. The upper left panel presents the spatial covariance structure of the leading mode, the upper right indicates the variance associated with 20 leading modes and the lower panel shows the leading PC where each weight represents a different member of the multi-model ensemble. The black symbols represent the ERA5 reanalysis, whereas red and blue curves mark CMIP5 and CMIP6 weights, respectively. The units are dimensionless in upper left and lower panels, as the final numbers are a product between the eigenpattern, eigenvalue and PC according to equation 1. The units of upper right panel is %.

**Table 1.** GCMs ranked according to their RMSE score for their TAS mean seasonal cycle common EOF results. A Wilcoxon rank sum test with continuity correction data gave W = 533.5, p-value = 0.03892 for the alternative hypothesis that the true location shift is less than 0. See the Supporting material for details behind the calculations.

| | | | | | |
|---|---|---|---|---|---|
| NorESM2-MM.ssp245.r1i1p1f1 | 0.136 | NorESM1-M.rcp45.r1i1p1_1 | 0.145 | TaiESM1.ssp245.r1i1p1f1 | 0.145 |
| MRI-ESM2-0.ssp245.r1i1p1f1 | 0.149 | AWI-CM-1-1-MR.ssp245.r1i1p1f1 | 0.153 | CNRM-ESM2-1.ssp245.r1i1p1f2 | 0.153 |
| NorESM1-ME.rcp45.r1i1p1_1 | 0.154 | CNRM-CM6-1-HR.ssp245.r1i1p1f2 | 0.156 | NorESM2-LM.ssp245.r1i1p1f1 | 0.156 |
| EC-EARTH.rcp45.r1i1p1_1 | 0.159 | CNRM-CM6-1.ssp245.r1i1p1f2 | 0.161 | E3SM-1-1.ssp245.r1i1p1f1 | 0.161 |
| FIO-ESM-2-0.ssp245.r1i1p1f1 | 0.163 | CESM2-WACCM.ssp245.r1i1p1f1 | 0.164 | EC-Earth3-CC.ssp245.r1i1p1f1 | 0.165 |
| GISS-E2-1-H.ssp245.r1i1p1f2 | 0.168 | GFDL-CM3.rcp45.r1i1p1_1 | 0.169 | KIOST-ESM.ssp245.r1i1p1f1 | 0.17 |
| BNU-ESM.rcp45.r1i1p1_1 | 0.171 | GISS-E2-1-G.ssp245.r1i1p1f2 | 0.172 | EC-Earth3-Veg.ssp245.r1i1p1f1 | 0.173 |
| MCM-UA-1-0.ssp245.r1i1p1f2 | 0.173 | MIROC-ESM-CHEM.rcp45.r1i1p1_1 | 0.175 | MPI-ESM-MR.rcp45.r1i1p1_1 | 0.176 |
| CAMS-CSM1-0.ssp245.r1i1p1f1 | 0.176 | CESM1-BGC.rcp45.r1i1p1_1 | 0.18 | ACCESS1.3.rcp45.r1i1p1_1 | 0.185 |
| CESM1-CAM5.rcp45.r1i1p1_1 | 0.185 | GFDL-ESM2G.rcp45.r1i1p1_1 | 0.185 | FGOALS-f3-L.ssp245.r1i1p1f1 | 0.185 |
| CCSM4.rcp45.r1i1p1_1 | 0.186 | MIROC-ESM.rcp45.r1i1p1_1 | 0.186 | CNRM-CM5.rcp45.r1i1p1_1 | 0.188 |
| MPI-ESM-LR.rcp45.r1i1p1_1 | 0.188 | ACCESS-ESM1-5.ssp245.r1i1p1f1 | 0.19 | IITM-ESM.ssp245.r1i1p1f1 | 0.19 |
| ACCESS1-0.rcp45.r1i1p1_1 | 0.192 | EC-Earth3-Veg-LR.ssp245.r1i1p1f1 | 0.192 | FIO-ESM.rcp45.r1i1p1_1 | 0.193 |
| GISS-E2-H.rcp45.r1i1p1_1 | 0.193 | CIESM.ssp245.r1i1p1f1 | 0.197 | IPSL-CM6A-LR.ssp245.r1i1p1f1 | 0.197 |
| MIROC-ES2L.ssp245.r1i1p1f2 | 0.197 | GISS-E2-H-CC.rcp45.r1i1p1_1 | 0.198 | CanESM5.ssp245.r1i1p1f1 | 0.198 |
| UKESM1-0-LL.ssp245.r1i1p1f2 | 0.198 | GISS-E2-R.rcp45.r1i1p1_1 | 0.199 | ACCESS-CM2.ssp245.r1i1p1f1 | 0.199 |
| EC-Earth3.ssp245.r1i1p1f1 | 0.199 | CMCC-CMS.rcp45.r1i1p1_1 | 0.2 | GISS-E2-R-CC.rcp45.r1i1p1_1 | 0.2 |
| GFDL-ESM2M.rcp45.r1i1p1_1 | 0.201 | IPSL-CM5A-MR.rcp45.r1i1p1_1 | 0.201 | MPI-ESM1-2-LR.ssp245.r1i1p1f1 | 0.202 |
| MIROC6.ssp245.r1i1p1f1 | 0.203 | CMCC-CM2-SR5.ssp245.r1i1p1f1 | 0.206 | IPSL-CM5A-LR.rcp45.r1i1p1_1 | 0.207 |
| MIROC5.rcp45.r1i1p1_1 | 0.209 | FGOALS-g3.ssp245.r1i1p1f1 | 0.21 | INM-CM4-8.ssp245.r1i1p1f1 | 0.211 |
| MRI-CGCM3.rcp45.r1i1p1_1 | 0.216 | CMCC-ESM2.ssp245.r1i1p1f1 | 0.216 | bcc-csm1-1.rcp45.r1i1p1_1 | 0.219 |
| HadGEM3-GC31-LL.ssp245.r1i1p1f3 | 0.22 | INM-CM5-0.ssp245.r1i1p1f1 | 0.221 | FGOALS.g2.rcp45_r1 | 0.227 |
| IPSL-CM5B-LR.rcp45.r1i1p1_1 | 0.23 | NESM3.ssp245.r1i1p1f1 | 0.232 | BCC-CSM2-MR.ssp245.r1i1p1f1 | 0.234 |
| HadGEM2-CC.rcp45.r1i1p1_1 | 0.237 | HadGEM2-ES.rcp45.r1i1p1_1 | 0.237 | KACE-1-0-G.ssp245.r1i1p1f1 | 0.237 |
| bcc-csm1-1-m.rcp45.r1i1p1_1 | 0.243 | CSIRO-Mk3-6-0.rcp45.r1i1p1_1 | 0.271 | CanESM2.rcp45.r1i1p1_1 | 0.278 |

**Table 2.** GCMs ranked according to their RMSE score for their PR mean seasonal cycle common EOF results. A Wilcoxon rank sum test with continuity correction data returned W = 303.5, and a p-value = 0.0001545 for the alternative hypothesis that the true location shift is less than 0.

| | | | | | |
|---|---|---|---|---|---|
| CNRM-CM6-1-HR.ssp245.r1i1p1f2 | 0.137 | EC-Earth3.ssp245.r1i1p1f1 | 0.139 | EC-Earth3-CC.ssp245.r1i1p1f1 | 0.142 |
| MRI-ESM2-0.ssp245.r1i1p1f1 | 0.143 | CMCC-ESM2.ssp245.r1i1p1f1 | 0.144 | CMCC-CM2-SR5.ssp245.r1i1p1f1 | 0.149 |
| EC-Earth3-Veg.ssp245.r1i1p1f1 | 0.153 | EC-Earth3-Veg-LR.ssp245.r1i1p1f1 | 0.158 | EC-EARTH.rcp45.r1i1p1_1 | 0.164 |
| IPSL-CM6A-LR.ssp245.r1i1p1f1 | 0.165 | CNRM-CM6-1.ssp245.r1i1p1f2 | 0.166 | MPI-ESM-LR.rcp45.r1i1p1_1 | 0.167 |
| KIOST-ESM.ssp245.r1i1p1f1 | 0.167 | UKESM1-0-LL.ssp245.r1i1p1f2 | 0.168 | AWI-CM-1-1-MR.ssp245.r1i1p1f1 | 0.17 |
| NorESM2-MM.ssp245.r1i1p1f1 | 0.17 | HadGEM3-GC31-LL.ssp245.r1i1p1f3 | 0.171 | CNRM-ESM2-1.ssp245.r1i1p1f2 | 0.173 |
| GFDL-ESM2G.rcp45.r1i1p1_1 | 0.174 | MIROC6.ssp245.r1i1p1f1 | 0.178 | MPI-ESM-MR.rcp45.r1i1p1_1 | 0.179 |
| CESM2-WACCM.ssp245.r1i1p1f1 | 0.179 | MRI-CGCM3.rcp45.r1i1p1_1 | 0.18 | FIO-ESM-2-0.ssp245.r1i1p1f1 | 0.18 |
| CESM1-BGC.rcp45.r1i1p1_1 | 0.181 | CESM1-CAM5.rcp45.r1i1p1_1 | 0.181 | CMCC-CMS.rcp45.r1i1p1_1 | 0.181 |
| CIESM.ssp245.r1i1p1f1 | 0.181 | CCSM4.rcp45.r1i1p1_1 | 0.182 | GFDL-CM3.rcp45.r1i1p1_1 | 0.182 |
| NorESM1-ME.rcp45.r1i1p1_1 | 0.182 | GISS-E2-H-CC.rcp45.r1i1p1_1 | 0.183 | NorESM1-M.rcp45.r1i1p1_1 | 0.183 |
| INM-CM5-0.ssp245.r1i1p1f1 | 0.184 | NorESM2-LM.ssp245.r1i1p1f1 | 0.184 | MCM-UA-1-0.ssp245.r1i1p1f2 | 0.185 |
| TaiESM1.ssp245.r1i1p1f1 | 0.186 | GISS-E2-1-G.ssp245.r1i1p1f2 | 0.187 | ACCESS-CM2.ssp245.r1i1p1f1 | 0.188 |
| MPI-ESM1-2-LR.ssp245.r1i1p1f1 | 0.188 | ACCESS1-0.rcp45.r1i1p1_1 | 0.189 | HadGEM2-ES.rcp45.r1i1p1_1 | 0.189 |
| BCC-CSM2-MR.ssp245.r1i1p1f1 | 0.189 | CAMS-CSM1-0.ssp245.r1i1p1f1 | 0.189 | E3SM-1-1.ssp245.r1i1p1f1 | 0.189 |
| NESM3.ssp245.r1i1p1f1 | 0.19 | INM-CM4-8.ssp245.r1i1p1f1 | 0.191 | CNRM-CM5.rcp45.r1i1p1_1 | 0.192 |
| GISS-E2-R.rcp45.r1i1p1_1 | 0.192 | CanESM5.ssp245.r1i1p1f1 | 0.192 | FGOALS.g2.rcp45_r1 | 0.193 |
| KACE-1-0-G.ssp245.r1i1p1f1 | 0.194 | bcc-csm1-1-m.rcp45.r1i1p1_1 | 0.196 | HadGEM2-CC.rcp45.r1i1p1_1 | 0.196 |
| bcc-csm1-1.rcp45.r1i1p1_1 | 0.197 | CSIRO-Mk3-6-0.rcp45.r1i1p1_1 | 0.199 | GFDL-ESM2M.rcp45.r1i1p1_1 | 0.199 |
| FGOALS-f3-L.ssp245.r1i1p1f1 | 0.199 | GISS-E2-R-CC.rcp45.r1i1p1_1 | 0.2 | MIROC-ESM-CHEM.rcp45.r1i1p1_1 | 0.201 |
| MIROC-ESM.rcp45.r1i1p1_1 | 0.202 | ACCESS-ESM1-5.ssp245.r1i1p1f1 | 0.202 | MIROC5.rcp45.r1i1p1_1 | 0.203 |
| ACCESS1.3.rcp45.r1i1p1_1 | 0.206 | MIROC-ES2L.ssp245.r1i1p1f2 | 0.209 | IPSL-CM5A-MR.rcp45.r1i1p1_1 | 0.212 |
| IPSL-CM5A-LR.rcp45.r1i1p1_1 | 0.22 | CanESM2.rcp45.r1i1p1_1 | 0.224 | FIO-ESM.rcp45.r1i1p1_1 | 0.232 |
| BNU-ESM.rcp45.r1i1p1_1 | 0.24 | | | | |

**Table 3.** GCMs ranked according to their RMSE score for their PSL mean seasonal cycle common EOF results. A Wilcoxon rank sum test with continuity correction data gave W = 296, with p-value = $3.8 \times 10^{-5}$ for the alternative hypothesis that the true location shift is less than 0.

| | | | | | |
|---|---|---|---|---|---|
| UKESM1-0-LL.ssp245.r1i1p1f2 | 0.15 | EC-Earth3-CC.ssp245.r1i1p1f1 | 0.152 | CNRM-ESM2-1.ssp245.r1i1p1f2 | 0.153 |
| HadGEM3-GC31-LL.ssp245.r1i1p1f3 | 0.153 | MRI-ESM2-0.ssp245.r1i1p1f1 | 0.157 | CNRM-CM6-1.ssp245.r1i1p1f2 | 0.158 |
| EC-EARTH.rcp45.r1i1p1_1 | 0.16 | CAMS-CSM1-0.ssp245.r1i1p1f1 | 0.165 | ACCESS-CM2.ssp245.r1i1p1f1 | 0.171 |
| CNRM-CM6-1-HR.ssp245.r1i1p1f2 | 0.173 | TaiESM1.ssp245.r1i1p1f1 | 0.173 | CESM2-WACCM.ssp245.r1i1p1f1 | 0.174 |
| EC-Earth3.ssp245.r1i1p1f1 | 0.174 | CNRM-CM5.rcp45.r1i1p1_1 | 0.175 | NorESM2-MM.ssp245.r1i1p1f1 | 0.175 |
| EC-Earth3-Veg-LR.ssp245.r1i1p1f1 | 0.176 | FIO-ESM-2-0.ssp245.r1i1p1f1 | 0.178 | CMCC-CMS.rcp45.r1i1p1_1 | 0.179 |
| AWI-CM-1-1-MR.ssp245.r1i1p1f1 | 0.179 | KIOST-ESM.ssp245.r1i1p1f1 | 0.179 | EC-Earth3-Veg.ssp245.r1i1p1f1 | 0.181 |
| MPI-ESM-MR.rcp45.r1i1p1_1 | 0.182 | BCC-CSM2-MR.ssp245.r1i1p1f1 | 0.184 | CIESM.ssp245.r1i1p1f1 | 0.184 |
| FGOALS-f3-L.ssp245.r1i1p1f1 | 0.186 | MPI-ESM-LR.rcp45.r1i1p1_1 | 0.187 | ACCESS-ESM1-5.ssp245.r1i1p1f1 | 0.188 |
| INM-CM4-8.ssp245.r1i1p1f1 | 0.188 | ACCESS1-0.rcp45.r1i1p1_1 | 0.189 | CESM1-CAM5.rcp45.r1i1p1_1 | 0.189 |
| NorESM1-M.rcp45.r1i1p1_1 | 0.189 | GISS-E2-H-CC.rcp45.r1i1p1_1 | 0.19 | NorESM1-ME.rcp45.r1i1p1_1 | 0.19 |
| KACE-1-0-G.ssp245.r1i1p1f1 | 0.19 | MIROC-ESM-CHEM.rcp45.r1i1p1_1 | 0.191 | CCSM4.rcp45.r1i1p1_1 | 0.193 |
| IPSL-CM6A-LR.ssp245.r1i1p1f1 | 0.193 | GFDL-CM3.rcp45.r1i1p1_1 | 0.194 | GISS-E2-H.rcp45.r1i1p1_1 | 0.194 |
| INM-CM5-0.ssp245.r1i1p1f1 | 0.196 | GISS-E2-1-G.ssp245.r1i1p1f2 | 0.197 | CESM1-BGC.rcp45.r1i1p1_1 | 0.199 |
| IITM-ESM.ssp245.r1i1p1f1 | 0.199 | GFDL-ESM2G.rcp45.r1i1p1_1 | 0.2 | CanESM5.ssp245.r1i1p1f1 | 0.2 |
| CMCC-CM2-SR5.ssp245.r1i1p1f1 | 0.2 | NorESM2-LM.ssp245.r1i1p1f1 | 0.2 | IPSL-CM5A-MR.rcp45.r1i1p1_1 | 0.201 |
| MIROC-ESM.rcp45.r1i1p1_1 | 0.201 | MPI-ESM1-2-LR.ssp245.r1i1p1f1 | 0.201 | CMCC-ESM2.ssp245.r1i1p1f1 | 0.203 |
| IPSL-CM5A-LR.rcp45.r1i1p1_1 | 0.204 | GISS-E2-R.rcp45.r1i1p1_1 | 0.207 | GISS-E2-R-CC.rcp45.r1i1p1_1 | 0.209 |
| NESM3.ssp245.r1i1p1f1 | 0.209 | FIO-ESM.rcp45.r1i1p1_1 | 0.21 | GFDL-ESM2M.rcp45.r1i1p1_1 | 0.21 |
| ACCESS1.3.rcp45.r1i1p1_1 | 0.212 | MIROC6.ssp245.r1i1p1f1 | 0.212 | HadGEM2-CC.rcp45.r1i1p1_1 | 0.213 |
| BNU-ESM.rcp45.r1i1p1_1 | 0.215 | CSIRO-Mk3-6-0.rcp45.r1i1p1_1 | 0.216 | MIROC5.rcp45.r1i1p1_1 | 0.217 |
| bcc-csm1-1-m.rcp45.r1i1p1_1 | 0.218 | HadGEM2-ES.rcp45.r1i1p1_1 | 0.22 | MRI-CGCM3.rcp45.r1i1p1_1 | 0.221 |
| CanESM2.rcp45.r1i1p1_1 | 0.223 | bcc-csm1-1.rcp45.r1i1p1_1 | 0.233 | MCM-UA-1-0.ssp245.r1i1p1f2 | 0.234 |
| MIROC-ES2L.ssp245.r1i1p1f2 | 0.235 | IPSL-CM5B-LR.rcp45.r1i1p1_1 | 0.244 | FGOALS.g2.rcp45_r1 | 0.246 |