# Peer review of "Various ways of using Empirical Orthogonal Functions for Climate Model evaluation"

_EGUsphere, 2022_

## Community Comment (CC2)

**EGUsphere-2022-1385 response to reviewer 2**

https://egusphere.copernicus.org/preprints/2023/egusphere-2022-1385/

**Major comments**

The presented EOF technique does not pursue the same target as the original common EOFs introduced by Flury (1984) and recently applied by Hannachi et al. (2022). These common EOFs find a set of basis vectors that approximately maximizes the variance within a number of datasets (each from one projection) simultaneously. The EOFs proposed by Benestad et al. find the exact EOFs that maximize the mixture of within variances (within each projection) and between cross-covariances (cross-covariance between all pairs of projections) of the combined dataset. As these are two different kinds of optimization, I would suggest to **find a new name for the presented technique**.

Besides, it would surely be very helpful to discuss the implications of this difference on the meaning of the resulting EOFs. The technique could further be **contrasted to multivariate EOFs, where all monthly values of any one projection are stacked in space-direction instead of time-direction**, as was done by Sanderson et al. (2015) to represent the similarity between projections in a joint multidimensional space, and/or to one of the many **tensor decompositions, where the space-time matrices of the individual projections are stacked along a third (model) direction**, and components are found, which present the main variations along those three directions (e.g. Cichocki et al., 2015). Aside from that, there exists a vast and slightly chaotic literature on multigroup/ multiblock/multitable PCA methods, which is also concerned with contriving ways to combine datasets of differing origin in one joint PCA.

The discussion of the presented applications seems, at least in my opinion, very much centered on the "easy" cases, where the variance is represented overwhelmingly by the first EOF and the reference lies well in the middle of the ensemble (seasonal cycles, interannual variability of TAS and PSL, and TAS trend). The authors could take the more complicated situation in the interannual variation of PR as an opportunity to **elaborate further on the inclusion of more than one PC in the subsequent evaluation, and on the possibility and consequences of the reference not falling inside the ensemble**.

I further suggest to consider a variation of the proposed method: to include only the ensemble of projections in the EOF analysis, and to project the reference, which might also be more than one references, onto the EOFs afterwards. It would appear to me as a "cleaner" approach not to meddle projections and references. On the other hand, in such

vast ensembles as used here, the exclusion of the reanalysis will hardly make any appreciable difference.

Sanderson, B.M., Knutti, R., Caldwell, P. (2015): A representative democracy to reduce interdependency in a multimodel ensemble. Journal of Climate, 28(13), pp. 5171-5194

Cichocki, A., Mandic, D., De Lathauwer, L., ... (2015): Tensor decompositions for signal processing applications: From two-way to multiway component analysis. IEEE Signal Processing Magazine, 32(2),7038247, pp. 145-163

**Thank you for your comments. The point about using a different name is well-taken, and the term 'common EOF' in our opinion is not optimal either, but was used by Barnett (1999)[1] who applied an ordinary EOF analysis to a joint dataset, just as we did here. Also, we have used this approach since 2001 as a framework for representing predictors in empirical-statistical downscaling, albeit for reanalysis-GCM pairs where the reanalysis has been used for calibration and GCM for projection. Nevertheless, this point is well-taken and will be included in the revised paper, as will the contribution to the body of knowledge represented by the two papers. Thanks for these references.**

**The suggested variation of our analysis is interesting, and we will discuss it in the revised paper. We nevertheless want to stick to our original approach, which also was used in Barnett (1999), as the motivation behind this study is the use of the common EOFs as predictors in empirical-statistical downscaling where they also have a role in the evaluation of both GCMs and the downscaled results. Furthermore, our hypothesis is that the subselection of ERA5 and the GCM data represent the same variable, region and time period, and hence the same statistics.**

**The point concerning a more complicated situation in the interannual variation is also an important one. The revised paper will discuss further the option of including more than one PC in the evaluation. And there is always a possibility that the reference does not fall inside the ensemble spread for a given PC, which would indicate that they belong to different statistical populations. This matters most for PCs representing a large fraction of variance, but can be ignored for high-order PCs associated with negligible variance that represent "numeric noise".**

Minor comments

Line 235 ff.: Is there any evidence that supports the proposition of independence between ensemble members? The QQ-plot suggests Gaussian distribution, but to my knowledge the QQ-plot gives no hint concerning independence. Furthermore, this would contradict the widely-recognized notion of strong interdependence between climate models.
* * *
[1] https://shorturl.at/dqxAG

From a physical point of view, we know that these models reproduce chaotic and stochastic variability on decadal scales and this is especially apparent if the ensemble is made up of simulations with one common model (Deser et al., 2012). For multi-model ensembles, there is also an additional component: model differences.

We should indeed expect a strong interdependence between climate models since they are built to represent the same physical system - what we also desire is that the aspects that are not well-established and uncertain should involve different choices/methods so that they also provide a decent sample of the parameter space of unknowns. But in practice, different groups often copy others' attempts so that model uncertainties are not so well sampled. Nevertheless, the simulated stochastic/chaotic regional internal variability appears to be more pronounced so that these concerns are secondary in this case.

PC plots in all figures: I find it difficult to distinguish the black curve (reanalysis) against the background of the dark blue curves (CMIP6).

Thanks for pointing this out - the clarity of the figure will be improved in the revised paper.

References:

Deser, C., Knutti, R., Solomon, S., and Phillips, A. S.: Communication of the role of natural variability in future North American climate, Nature Climate Change, 2, 775–779, 2012, DOI: 10.1038/NCLIMATE1562

---

## Author Response (AR1)

**EGUsphere-2022-1385 response to reviewer 1**

https://egusphere.copernicus.org/preprints/2023/egusphere-2022-1385/

Major Comments
* * *
The SVD-based common EOFs method used in the paper is akin to the combined EOFs (e.g., Navarra and Simoricini 2010) where the different datasets are packed in one single large array, which is then analysed via SVD. Of course the difference is in the way the data bloc matrices are arranged in the large array. The result is a set of individual eigenelements (i.e. EOF in S-mode as in Barnett (1998) and also here, or PC in T-mode as in combined PCA, see, e.g. Jolliffe (2002)) associated with corresponding eigenvalues. The original common EOFs method as presented first by Flury (1984, 1986), see also Hannachi (2021) for earlier literature, analyses different covariance matrices, for which one common EOF has different explained variances depending on the data (or model run). Clearly this version gives more degrees-of-freedom to the common EOFs compared to the one defined by Barnett (1998) or here where one common EOF has one single explained variance for all the models' simulations. One benefit of the former is that these eigenvalues --for one given common EOF-- can be made useful to weigh the different models, and can be used in various other ways, e.g., to get the models' climatology. In addition, it overcomes the issue of scaling in the different datasets.

Of course I must say though that the SVD-based common EOFs (Barnett 1998 and the present manuscript) is computationally much faster and is convenient for application with large number of GCMs runs as in this paper. I think these points, with the above references highlighting the historical context of common EOF/PCs should be included in the revised version. In Hannachi et al. (2022) the references we mentioned there are more related to climate research. Some other references (e.g., Barnett 1999) were missing because the search engine did not find them as they do not mention common EOFs/PCs in the title. In any case, the first time common EOF/PCs was mentioned was in Flury (1984).

**Thanks for this valuable comment and the additional list of references - it's indeed important to acknowledge relevant work and give credit to past scientific effort. The cited work will be included in the revised final paper.**

Minor Comments
* * *
Pg 3 - Please change TAS to SAT (surface air temperature) and PSL to SLP (sea level pressure) in page 3 and elsewhere. **- I understand the preference to use SAT and SLP, however, we thought it still may be better to use the standard variable names from the CMIP archive.**

Pg 3, l71: 'vector' --> 'value' **- changed. Thanks for spotting this mistake.**

Pg 6, near l171, l175 - repetition. **Thanks - the text has been revised.**

Fig 5, top panel: y-axis label: add "Relative (or scaled) rank". **I am unfamiliar with the term 'relative rank', but think that the word 'rank' is in line with Ranking in statistics as described in https://en.wikipedia.org/wiki/Ranking**

Pg 8, l240 - ensemble spread cannot be normal - could be truncated normal perhaps. **Fig. 8 suggests that the ensemble distribution is approximately normal except for some exceptions. But, why can't ensemble spread be normal - because the ensemble members are bounded random variables from either below or above?**

Fig. 8, I presume there is one value per model, right? Is it global mean of the climatology? **Yes, and the points represent the mean for a given season. The revised paper will make this clearer.**

Fig. 9, top left and bottom panels, units: oC/yr **- °C/decade, thanks for pointing this out. The revised paper will make this clearer.**

References
* * *
Flury B.N., Common principal components in k groups. J. Am. Stat. Assoc., 1984.

Flury B.N., and W. Gutchi: An algorithm for simultaneous orthogonal transformation of several positive definite symmetric matrices to nearly diagonal form. SIAM J. Sci. Stat. Comput., 1986.

Hannachi, A., Pattern Identification and Data Mining in Weather and Climate, Springer Nature, 2021.

Jolliffe I.T., Principal Component Analysis, Second Edition, Springer Nature, 2002.

Navarra, A., and V. Simoncini, A Guide to Empirical Orthogonal Functions for Climate Data Analysis. Springer, 2010.

**EGUsphere-2022-1385 response to reviewer 2**

https://egusphere.copernicus.org/preprints/2023/egusphere-2022-1385/

**Major comments**

The presented EOF technique does not pursue the same target as the original common EOFs introduced by Flury (1984) and recently applied by Hannachi et al. (2022). These common EOFs find a set of basis vectors that approximately maximizes the variance within a number of datasets (each from one projection) simultaneously. The EOFs proposed by Benestad et al. find the exact EOFs that maximize the mixture of within variances (within each projection) and between cross-covariances (cross-covariance between all pairs of projections) of the combined dataset. As these are two different kinds of optimization, I would suggest to **find a new name for the presented technique**.

Besides, it would surely be very helpful to discuss the implications of this difference on the meaning of the resulting EOFs. The technique could further be **contrasted to multivariate EOFs, where all monthly values of any one projection are stacked in space-direction instead of time-direction**, as was done by Sanderson et al. (2015) to represent the similarity between projections in a joint multidimensional space, and/or to one of the many **tensor decompositions, where the space-time matrices of the individual projections are stacked along a third (model) direction**, and components are found, which present the main variations along those three directions (e.g. Cichocki et al., 2015). Aside from that, there exists a vast and slightly chaotic literature on multigroup/ multiblock/multitable PCA methods, which is also concerned with contriving ways to combine datasets of differing origin in one joint PCA.

The discussion of the presented applications seems, at least in my opinion, very much centered on the "easy" cases, where the variance is represented overwhelmingly by the first EOF and the reference lies well in the middle of the ensemble (seasonal cycles, interannual variability of TAS and PSL, and TAS trend). The authors could take the more complicated situation in the interannual variation of PR as an opportunity to **elaborate further on the inclusion of more than one PC in the subsequent evaluation, and on the possibility and consequences of the reference not falling inside the ensemble**.

I further suggest to consider a variation of the proposed method: to include only the ensemble of projections in the EOF analysis, and to project the reference, which might also be more than one references, onto the EOFs afterwards. It would appear to me as a

"cleaner" approach not to meddle projections and references. On the other hand, in such vast ensembles as used here, the exclusion of the reanalysis will hardly make any appreciable difference.

Sanderson, B.M., Knutti, R., Caldwell, P. (2015): A representative democracy to reduce interdependency in a multimodel ensemble. Journal of Climate, 28(13), pp. 5171-5194

Cichocki, A., Mandic, D., De Lathauwer, L., ... (2015): Tensor decompositions for signal processing applications: From two-way to multiway component analysis. IEEE Signal Processing Magazine, 32(2),7038247, pp. 145-163

**Thank you for your comments. The point about using a different name is well-taken, and the term 'common EOF' in our opinion is not optimal either, but was used by Barnett (1999)[1] who applied an ordinary EOF analysis to a joint dataset, just as we did here. Also, we have used this approach since 2001 as a framework for representing predictors in empirical-statistical downscaling, albeit for reanalysis-GCM pairs where the reanalysis has been used for calibration and GCM for projection. Nevertheless, this point is well-taken and will be included in the revised paper, as will the contribution to the body of knowledge represented by the two papers. Thanks for these references.**

**The suggested variation of our analysis is interesting, and we will discuss it in the revised paper. We nevertheless want to stick to our original approach, which also was used in Barnett (1999), as the motivation behind this study is the use of the common EOFs as predictors in empirical-statistical downscaling where they also have a role in the evaluation of both GCMs and the downscaled results. Furthermore, our hypothesis is that the subselection of ERA5 and the GCM data represent the same variable, region and time period, and hence the same statistics.**

**The point concerning a more complicated situation in the interannual variation is also an important one. The revised paper will discuss further the option of including more than one PC in the evaluation. And there is always a possibility that the reference does not fall inside the ensemble spread for a given PC, which would indicate that they belong to different statistical populations. This matters most for PCs representing a large fraction of variance, but can be ignored for high-order PCs associated with negligible variance that represent "numeric noise".**

Minor comments

Line 235 ff.: Is there any evidence that supports the proposition of independence between ensemble members? The QQ-plot suggests Gaussian distribution, but to my knowledge the
* * *
[1] https://shorturl.at/dqxAG

QQ-plot gives no hint concerning independence. Furthermore, this would contradict the widely-recognized notion of strong interdependence between climate models.

From a physical point of view, we know that these models reproduce chaotic and stochastic variability on decadal scales and this is especially apparent if the ensemble is made up of simulations with one common model (Deser et al., 2012). For multi-model ensembles, there is also an additional component: model differences.

We should indeed expect a strong interdependence between climate models since they are built to represent the same physical system - what we also desire is that the aspects that are not well-established and uncertain should involve different choices/methods so that they also provide a decent sample of the parameter space of unknowns. But in practice, different groups often copy others' attempts so that model uncertainties are not so well sampled. Nevertheless, the simulated stochastic/chaotic regional internal variability appears to be more pronounced so that these concerns are secondary in this case.

PC plots in all figures: I find it difficult to distinguish the black curve (reanalysis) against the background of the dark blue curves (CMIP6).

Thanks for pointing this out - the clarity of the figure will be improved in the revised paper.

References:

Deser, C., Knutti, R., Solomon, S., and Phillips, A. S.: Communication of the role of natural variability in future North American climate, Nature Climate Change, 2, 775–779, 2012, DOI: 10.1038/NCLIMATE1562